# Mechanism of Synergistic Photoinactivation Utilizing Curcumin and Lauric Arginate Ethyl Ester against *Escherichia coli* and *Listeria innocua*

**DOI:** 10.3390/foods12234195

**Published:** 2023-11-21

**Authors:** Victor Ryu, Joseph Uknalis, Maria G. Corradini, Piyanan Chuesiang, Lynne McLandsborough, Helen Ngo, Tony Jin, Xuetong Fan

**Affiliations:** 1United States Department of Agriculture, Agricultural Research Service, Eastern Regional Research Center, 600 East Mermaid Lane, Wyndmoor, PA 19038, USA; victor.ryu@usda.gov (V.R.); joseph.uknalis@usda.gov (J.U.); helen.ngo@usda.gov (H.N.); tony.jin@usda.gov (T.J.); 2Food Science Department & Arrell Food Institute, University of Guelph, Guelph, ON N1G 2W1, Canada; mcorradi@uoguelph.ca; 3Department of Food Technology, Faculty of Science, Chulalongkorn University, Bangkok 10330, Thailand; pchuesiang@gmail.com; 4Department of Food Science, University of Massachusetts Amherst, Amherst, MA 01003, USA; lm@foodsci.umass.edu

**Keywords:** microbial photoinactivation, curcumin, lauric arginate ethyl ester, photosensitizer, reactive oxygen species

## Abstract

This study investigated the mechanism of how lauric arginate ethyl ester (LAE) improves the photoinactivation of bacteria by curcumin after diluting the 100 µmol/L stock curcumin-LAE micelle solution to the concentration used during the treatment based on the curcumin concentration. The photoinactivation of bacteria was conducted by irradiating the 1 µmol/L curcumin-LAE solution containing cocktails of *Escherichia coli* and *Listeria innocua* strains (7 log CFU/mL) for 5 min with UV-A light (λ = 365 nm). The changes in solution turbidity, curcumin stability, and bacterial morphology, viability, and recovery were observed using SEM, TEM, and live/dead cell assays. The study found that LAE enhances the photoinactivation of bacteria by increasing the permeability of cell membranes which could promote the interaction of reactive oxygen species produced by photosensitized curcumin with the cell components. The combination of curcumin and LAE was demonstrated to be more effective in inhibiting bacterial recovery at pH 3.5 for *E. coli*, while LAE alone was more effective at pH 7.0 for *L. innocua*.

## 1. Introduction

Photodynamic therapy (PDT) uses a combination of light and a photosensitizing agent [1]. Recently, PDT has also been recognized as a promising microbial inactivation strategy with several applications [1]. As such, photodynamic inactivation (PDI) derived from PDT can be applied in the sanitation of food or food surfaces. The food industry uses approved sanitizers to decrease the number of microorganisms that may be present on a surface or in final food products. However, conventional sanitizers, which are poorly biodegradable, can cause severe corrosion, and produce harmful by-products [2,3]. Furthermore, inadequate sanitation and biofilm formation can result in the emergence of bacteria that develop resistance to sanitizers, which may subsequently lead to antibiotic resistance [3,4]. Tong et al. found that exposing *Pseudomonas* sp. to sub-minimal inhibitory concentrations (MIC) of sodium hypochlorite increased their resistance to antibiotics. This was attributed to the increased production of antioxidant enzymes after exposure to sodium hypochlorite and the upregulation of genes related to the SOS response, efflux system, and antibiotic resistance enzymes [5]. Khan, Beattie, and Knapp also found bacteria that survived in chlorinated drinking water had a higher chance of developing antibiotic resistance and were more likely to withstand disinfection with chlorine [6].

PDI is one way to replace conventional sanitation methods used in the food industry. It combines photosensitizers (PSs) and light at a range of wavelength. PSs can absorb light and cause a redox reaction with surrounding compounds, such as ground state oxygen, turning them into reactive oxygen species (ROS). The non-selective nature of ROS in targeting necessary molecules for bacteria’s survival has led several studies to conclude that bacteria are unlikely to obtain resistance against them [7,8]. Bacteria can produce superoxide dismutase or catalase to disproportionate hydrogen peroxide or superoxide anions, respectively. However, the emergence of resistant bacteria toward ROS is difficult due to their inability to produce enzymes that can neutralize the harmful effects of the hydroxyl radical (^●^OH) and singlet oxygen (^1^O_2_) [7]. 

Curcumin, a yellow pigment found in plants of the *Curcuma longa* species, was used as a food-grade PS in this study. Curcumin is also the primary curcuminoid extracted from turmeric, which is indigenous to Southeast Asia. It has numerous reported health benefits and has been used medicinally for nearly 4000 years. In the food industry, curcumin is utilized as a food colorant and flavoring agent as it is generally recognized as safe (GRAS). The European Safety Authority states that the acceptable daily intake of curcumin is 3 mg/kg body weight per day [9]. However, curcumin is highly unstable to light exposure or solubilization in basic aqueous solutions, and nucleates (forming a precipitate) in acidic aqueous solutions [10]. To increase the stability and photoinactivation of curcumin in aqueous solutions, our previous study utilized a micellar system composed of lauric arginate ethyl ester (LAE) [11]. LAE is a GRAS cationic surfactant, known for its antimicrobial activity in foods. Its primary mode of action is to disrupt the bacterial cell membrane, causing the aggregation of DNA without lysis, leakage of inner cellular materials, and accumulation of intracellular ROS [12]. The U.S. FDA has established regulations that permit the use of LAE as an additive in meat, poultry, cheese, etc., with a maximum allowable concentration of 200 mg/kg [13]. In acidic aqueous solutions containing both LAE and curcumin, a synergistic effect on photoinactivation against a cocktail of *E. coli* was observed [11]. However, the exact inactivation mechanism(s) are unknown. 

This study investigated the mechanism of synergistic photoinactivation between curcumin and LAE. Our goal was to provide information on potential mechanisms of synergistic antimicrobial activity when both a PS and the selected surfactant, i.e., LAE, are present.

## 2. Materials and Methods

### 2.1. Materials

Curcumin with a purity greater than 97% was obtained from TCI Chemicals (C2302-5G, Montgomeryville, PA, USA). LAE was purchased from Ambeed (A621851, Arlington Hts, IL, USA). Propylene glycol was obtained from Millipore Sigma (P4347-500 mL, Burlington, MA, USA). Sodium chloride was used to prepare a 0.85% saline solution (S9888-500 g, Sigma-Aldrich,, Allentown, PA, USA). Citric acid and sodium citrate were used to make sodium citrate buffer (5 mmol/L) (C83155-500 g, Sigma-Aldrich, and 0754-12, Mallinckrodt chemicals, Hampton, NJ, USA); 200-proof ethanol was obtained from Koptec (64-17-5, King of Prussia, PA, USA); and 2.5% glutaraldehyde was obtained from Electron Microscopy Sciences [EMS] (Hatfield, PA, USA). The Live/Dead^TM^ BacLight^TM^ bacterial viability kit (L7012) was from Thermofisher (Thermo Scientific, Portsmouth, NH, USA). The 35 mm poly-D-lysine coated dish (P35GC-1.4-14-C) was from MatTek (Ashland, MA, USA).

### 2.2. Stock Curcumin-LAE Solution

A 4 mmol/L stock solution of curcumin was made using ethanol as a solvent, and a 10.5% *w*/*v* stock solution of LAE was prepared using propylene glycol. To create a 100 µmol/L stock solution of curcumin-LAE micelle, the LAE stock solution was first diluted in distilled water with pH 3.5 and agitated with the stir bar at 125 rpm for 15 min. The curcumin stock solution was then added to the LAE solution at a rate of 2.5 mL/min and stirred for 15 min producing a stock solution with an LAE concentration of 1.05% (*v*/*v*). This additional stirring was performed to monitor the formation of large curcumin crystals in the aqueous phase as described previously. After the additional stirring, it was filtered and sterilized using a 0.45 µm syringe filter (02915-22, Cole-Palmer, Vernon Hills, IL, USA) before storing at 20 °C. 

### 2.3. Stability of Encapsulated Curcumin after Dilution

The stability of the encapsulated curcumin after dilution was observed by diluting a 100 µmol/L stock curcumin-LAE micelle solution 1:100 using a 0.85% saline solution at pH 3.5. After dilution, 1 µmol/L curcumin-LAE solution was then transferred to quartz cuvettes with a 1 cm light path, and its turbidity was determined by measuring the absorbance at 600 nm using a UV–Visible spectrophotometer (Shimadzu Scientific, Columbia, MD, USA). Also, to determine its stability over time in solution, the absorbance at 400 nm was measured using a 1 nm slit at 10 min intervals for 1 h. In addition, the fluorescence intensity of the 1 µmol/L curcumin-LAE solution was obtained as well. The 1 µmol/L curcumin-LAE solution’s fluorescence emission spectra were determined using a spectrofluorometer (Shimadzu Scientific, Columbia, MD, USA). The excitation wavelength was 365 nm and the range of the emission wavelength was set from 390 to 800 nm, and both the excitation and emission slit size was fixed to 5 nm.

### 2.4. Bacterial Culture Conditions

Two separate cocktails of *E. coli* or *L. innocua* were prepared from three strains of *E. coli* O157:H7 (ATCC-700728), K12 (ATCC-23716), and Seattle 1946 (ATCC-25922), and three strains of *L. innocua* Seelinger (ATCC-43547, 33090, 51742) purchased from American Type Culture Collections (Manassas, VA, USA). The frozen stocks of *E. coli* and *L. innocua* cultures were made by mixing a cryopreservative solution (MicrobankTM 2D, Pro-Lab Diagnostics, Roundrock, TX, USA) and storing them in cryovials at −80 °C. For preparation of the working stock used during the experiment, the cultures were streak-plated onto the selective media, MacConkey Sorbitol Agar (279100, BD Diagnostic Systems, Berkshire, UK) or Polymyxin-acriflavine-LiCl-ceftazidime-aesculin-mannitol (PALCAM) agar (222530, BD Diagnostic Systems), and stored at 4 °C up to 2 weeks. Overnight cultures were prepared by inoculating a colony from the working stock into tryptic soy broth (TSB; 211825, BD Diagnostic Systems) and incubating it at 37 °C on a 125 rpm shaker for 18 h. For the photoinactivation assay, the optical density at 600 nm (OD600) was measured after 1:10 dilution of the 18 h cultures of *E. coli* and *L. innocua*. They were then adjusted to 0.15 cm^−1^ and 0.12 cm^−1^, respectively, to confirm that the bacterial concentration of the cultures was 8 log CFU/mL. A 0.85% saline solution was used to wash the bacteria by centrifugation at 2182 g for 10 min and resuspension. The *E. coli* and *L. innocua* cocktail was created by combining equal volumes of the washed cultures. Serial dilution and spread plating on Tryptic Soy Agar (TSA; 236920, Neogen, Lansing, MI, USA) were used to confirm the number of bacteria, with the aim of obtaining an initial bacterial concentration of 7 log CFU/mL for each treatment.

### 2.5. Bacterial Photoinactivation

Samples prepared for the photoinactivation assay contained 7 log CFU/mL of untreated bacteria as a control, and with 1 µmol/L curcumin-LAE solution or individual components of 1 µmol/L curcumin-LAE solution (1 µmol/L curcumin or 105 μg/mL LAE). Aliquots of 2 mL of these samples were pipetted to 4 wells of sterile, non-treated 24-well plates (Falcon^®^, 351147, Glendale, AZ, USA). The plates were then stored in the dark for 5 min. After this, the samples were divided into 2 groups, non-irradiated and irradiated. The former group was left in the dark for another 5 min, while the latter group was irradiated for 5 min using a CL-3000L-crosslinker (Analytik jena, Tewksberry, MA, USA) that has six 8 W UV-A lamps (λ = 365 nm). The distance between the samples in the 24-well plate and the UV-A lamps was fixed to 9 cm by placing it on a heightened platform inside the chamber. The location of the 4 wells was also adjusted to ensure equal exposure to an irradiance of 9–9.3 mW/cm^2^ for each well. Verification of the irradiance and temperature in the wells were performed using a UV A/B light meter (850009, Sper Scientific, Scottsdale, AZ, USA) and a digital thermometer (15-077-8, Fisher Scientific), respectively. After irradiation or non-irradiation, LAE was neutralized by diluting the sample in 5 mmol/L sodium citrate buffer at pH 3.5 or 5, as it was able to precipitate LAE [14]. The 5 mmol/L sodium citrate buffer used in this study did not influence the photoinactivated bacteria.

### 2.6. Bacterial Growth after Photoinactivation 

#### 2.6.1. Monitoring Growth Using a Plate Reader

The recovery of bacteria after photoinactivation was determined. First, 100 µL of the treated samples, diluted 1:10 in a corresponding 5 mmol/L sodium citrate buffer, was mixed with 100 µL of growth medium (TSB) in each well of a 96-well plate. This dilution in TSB prevented further photoinactivation by curcumin because the resulting solution was at pH 7 [15]. The OD600 of the samples was obtained using a microplate reader (Synergy H1, Biotek Instrument Inc., Santa Clara, CA, USA) every 1 h for 24 h at 37 °C with shaking for 30 s. The obtained data were exported using imager software (Gen 5, version 3.11, Biotek Instrument Inc., Santa Clara, CA, USA).

#### 2.6.2. Microbial Growth Modeling

The recovery of the *E. coli* and *L. innocua* cocktail after treatment was observed by measuring their OD600 at various time intervals using a microplate reader. The modified logistic model, as described in [16], was used to characterize the data:(1)$$Y\left(t\right)=\frac{a}{1+exp(k*\left(t-t_{c}\right))}-\frac{a}{1+exp(k*t_{c})}$$

The ratio of the momentary OD600 value to the initial OD600 value is represented by *Y(t)*. The asymptotic value of the growth curve is represented by “*a*”, the growth rate is represented by “*k*”, and the inflection point of the growth curve is represented by “*t_c_*”. The experimental data were fit with Equation (1) using a nonlinear regression routine in Mathematica 12.2 (Wolfram Research, Inc. Champaign, IL, USA). The accuracy of the fit was determined based on the mean squared error.

### 2.7. Scanning Electron Microscopy (SEM)

For SEM, acetone-cleaned 12 mm Micro-cover glass slides were used to adhere 50 µL of bacteria for 30 min. A total of 2 mL of 2.5% glutaraldehyde was added and allowed to fix for an additional 30 min. The samples were then rinsed two times with 2–3 mL of the 0.1 mol/L imidazole for 30 min, followed by rinsing with 50, 80, 90% ethanol solution for 30 min each, 2–3 mL. Before critical-point drying, the samples were then washed three times with 2 mL of 100% ethanol. The samples were placed in a critical point drying apparatus, and further dried using liquid carbon dioxide for 20 min. The samples were mounted and sputter gold coated for 1 min before subjecting to a FEI Quanta 200 F Scanning Electron Microscope (Hillsboro, OR, USA) with an accelerating voltage of 10 KV in high vacuum mode.

### 2.8. Transmission Electron Microscopy (TEM)

2.5% glutaraldehyde solution was used to fix the cells for 30 min, prior to centrifugation. The acquired pellet was resuspended in 10 μL of warm 1% agarose. This pellet was then washed twice with 0.1 mol/L imidazole solution for 30 min at room temperature. The pellet was fixed in the fume hood for 1 h after exposing it with 100 µL of a 1% osmium tetroxide (EMS) solution. The micropipette was used to resuspend the pellet and it was held for 1 h. The pellet was washed for 30 min with 1 mL 0.1 mol/L imidazole. Then it was dehydrated with graded 50, 80, and 90% ethanol solutions, for 30 min each. The sample was washed 3 times with 100% ethanol for 30 min each. Acetonitrile was then added to replace ethanol, twice for 5 min. Ladd LX-112 resin (Ladd Research, Williston, VT, USA) was mixed and infiltrated at 50%, 75%, 100% (2×) with acetonitrile. A vacuum oven at 60 and 25 °C in Hg overnight was used to cure the resin. A Reichert Ultracut S (Leica, Vienna, Austria) with a Diatome Ultra 45-degree diamond knife was used to cut the resin in sections with a 70 nm width. Sections were obtained on a copper 200 mesh grid and stained with a 1% solution of uranyl acetate, for approximately 1 min, rinsed with DI water, counterstained with Reynolds (1963) lead citrate for 1 min, and then rinsed with DI water. A Hitachi HT7800 TEM (Tokyo, Japan) with an accelerating voltage of 80 KV was used to observe the thin sections, and they were imaged with an AMT detector (Danvers, MA, USA). 

### 2.9. Live/Dead Cell Assay

The proportion of cells that had a permeable membrane was determined using a live/dead cell assay. A solution of 3 µL of the dye mixture with 1:1 ratio of Syto 9 and propidium iodide from the Live/Dead^TM^ BacLight^TM^ bacterial viability kit (L7012) was made in a centrifuge tube. Then, 1 mL of the bacterial suspension (7 log CFU/mL) was added. This mixture was then held in the dark for 15 min at ambient temperature. A total of 10 µL of the sample was placed on a 35 mm poly-D-lysine coated dish (P35GC-1.4-14-C). The fluorescence signal of the cell permeable dyes was obtained using a confocal microscope (Leica DMI 4000B, Wetzlar, Germany). When the focal plane was found with the microscope, a 488 nm laser was used to excite the dyes. The obtained micrographs were analyzed using Image J 1.53k software (National Institutes of Health, Bethesda, MD, USA) to determine the proportion of cells in the sample that had a permeable membrane as indicated by the emission wavelength from the propidium iodide.

### 2.10. Data Acquisition and Analysis

The experiments were conducted in triplicate. Statistical analyses were performed using IBM SPSS Statistics software (Version 16, IBM, Philadelphia, PA, USA) with a significance level of *p* ≤ 0.05.

## 3. Results and Discussion

### 3.1. Possible Mechanisms of Synergistic Photoinactivation

In the previous study, the synergistic photoactivated antimicrobial activity was observed between 1 μmol/L curcumin and 0.248 µmol/L LAE [11]. The dilution of a 100 µmol/L stock curcumin-LAE micelle solution to the treatment level of 1 µmol/L curcumin-LAE solution, based on the curcumin concentration, destabilized micelles, as the concentration of LAE present in the diluted solution is 0.248 µmol/L while LAE’s critical micelle concentration is around 5.7 μmol/L [14]. Even with the destabilization of the LAE micelle, synergistic antimicrobial activity between curcumin and LAE was observed. This could be explained based on three different phenomena. First, LAE could prevent curcumin from nucleation, allowing less light scattering in water and enabling light to excite curcumin molecules located further from the light source. Therefore, the absorbance of 1 μmol/L curcumin and 1 μmol/L curcumin-LAE solution at 600 nm was measured after 10 min in 0.85% saline solution. The solution turbidity showed no difference at the selected time points, which were chosen as they correspond to the completion of the photoinactivation assay (Appendix A). Second, the presence of surfactant could prevent curcumin from crystallizing over time in water and precipitating out of solution, which reduces the curcumin concentration in the solution and lowers the production of ROS. The presence of LAE stabilized curcumin in water, as indicated by the higher absorbance (400 nm) and fluorescence intensity (λ_exc_ = 365 nm, λ_em_ = 540 nm) over time compared to when curcumin was present in the solution alone (Appendix A). Dahl et al. observed that the rate of photoinactivation of different species of bacteria decreased when unbound or loosely bound curcumin was removed by discarding the bulk dye solution after preincubation [17]. Thus, the presence and amount of unbound molecular curcumin in the bulk solution could be a contributing factor to the synergistic photoinactivation. Lastly, LAE may compromise the integrity of the bacterial cell membrane, allowing for better partitioning of curcumin into the membrane and promoting better interactions between ROS and essential cellular components during irradiation. Ryu et al. identified a difference in curcumin localization, i.e., outside vs. inside a cell membrane, when a surfactant was present using fluorescence lifetime imaging microscopy [15]. Therefore, we will further discuss whether the membrane damage was caused by LAE during the photoinactivation in the consecutive sections based on SEM, TEM, and live/dead cell assay micrographs of treated cells.

### 3.2. Recovery of Cells after Photoinactivation

Cell recovery following photoinactivation treatment was monitored using a plate reader, and a modified logistic model (Equation (1)) was used to fit the bacterial growth curves. Treatment effectiveness was determined by the maximum level of growth (*a*). At pH 3.5, only the irradiated curcumin-LAE solution inhibited *E. coli* growth for 24 h, while after other treatments *E. coli* grew up to a similar maximum level of recovery (*a*) (Table 1, Figure 1). At pH 7, irradiated and non-irradiated curcumin-LAE solutions inhibited *E. coli* growth. Both irradiated and non-irradiated *E. coli*’s growth was inhibited, but the irradiated LAE solution effectively hindered growth, shown by differences in the maximum level of recovery (*a*). Photoinactivation by curcumin alone inhibited *L. innocua* at pH 3.5 for 24 h but not at pH 7, indicating curcumin’s greater antilisteria effectiveness at pH 3.5. When *L. innocua* was treated with LAE alone and irradiated in a pH 3.5 solution, the maximum level of recovery (*a*) was higher than when it was non-irradiated (Table 2, Figure 1). A previous study reported that when *E. coli* and *L. innocua* is treated with UV-A and LAE, it results in synergistic antimicrobial activity due to oxidative stress on the cells, which depended on whether the cells were metabolizing or not [18]. LAE was more effective at inhibiting both bacteria at pH 7 compared to pH 3.5, consistent with previous reports of limited LAE antimicrobial activity at an acidic pH for short treatment times (<20 min) [11]. However, the recovery data presented have shown that the subpopulation of bacteria was able to survive the treatment and could propagate when given enough time.

The cocktail of *E. coli* was more resistant toward treatment compared to the *L. innocua* cocktail. The trend in susceptibility to PDI was similar to that of a previous antimicrobial study, in which an *L. innocua* cocktail was more vulnerable than *E. coli* [11]. This could be attributed to the differences in Gram-positive and Gram-negative bacteria’s cell membrane structure [19]. The outer layer of Gram-positive bacteria is made of a thick coarse meshwork of a porous peptidoglycan layer, vulnerable to small molecules that could easily penetrate and bind with the cytoplasmic membrane [20,21]. Conversely, the outer membrane of *E. coli*, a Gram-negative bacteria, provides a tighter and more efficient barrier to neutral or hydrophobic compounds. Also, due to the negative charge on the cell membrane of Gram-negative bacteria induced by lipopolysaccharide, it prevents anionic compounds from binding with the cytoplasmic membrane [22]. The greater resistance of *E. coli* towards the photoactivated antimicrobial activity of curcumin, when used alone, compared to *L. innocua* may be due to a high proportion of the neutral form of curcumin molecules at pH 3.5 [10].

### 3.3. SEM Micrographs

The surface morphology of bacteria changed when different curcumin-LAE solutions or their individual components were used. For *E. coli*, LAE and curcumin-LAE solutions caused dimples on the cell membrane, indicating damaged surfaces and leakages of cellular material, similar to the micrographs reported previously (Figure 2) [20]. According to the cell recovery data, these damages were not lethal except when the treatment involved the curcumin-LAE solution at pH 3.5 or just LAE at pH 7 (Figure 1). For *L. innocua*, there were indications of cell leakages after irradiation with curcumin, LAE, and curcumin-LAE solutions at both pH 3.5 and 7 (Appendix A). However, the majority of cells appeared intact and exhibited less membrane damage than *E. coli* treated with LAE-containing solutions. LAE is a cationic surfactant with its pKa around 10–11 [23]. Hence, the arginine guanidium group of LAE carries a positive charge at all treatment pHs (3.5 and 7), which suggests that LAE can electrostatically interact with the negatively charged cell membranes of both Gram-positive and Gram-negative bacteria. Furthermore, a study has reported that the charge density of the representative Gram-negative bacteria, *E. coli*, had a seven-times-larger negative charge density than the Gram-positive bacteria *Lactobacillus rhamnosus* [24]. Therefore, the observed lesser damage on the outermost layer of *L. innocua* than *E. coli* might be due to the lower charge density of their cellular membrane which leads to less interaction between LAE and the membrane.

### 3.4. TEM Micrographs

The cell morphology of bacteria also changed when different components of curcumin-LAE solutions were individually used for treatment. For *E. coli*, all samples with LAE showed dissolution of the cell membrane. For samples with LAE at pH 3.5, some *E. coli* cells exhibited large white spots that could indicate either large membrane channels due to solubilization of the outer membrane or cytoplasmic collapse (Figure 3) [20,25]. At pH 7, *E. coli* treated with solutions including LAE caused the cytoplasmic membrane to exhibit intracytoplasmic coagulation and condensation, which could be seen by black clumps on the edge of the cell [25]. Coagulated material found in the cytoplasm might be a result of abnormal protein microprecipitation or the denaturation of membrane components [20]. For *L. innocua*, there were indications of cell leakages from the ones irradiated with curcumin and those containing LAE in their treatments at both pH 3.5 and 7 (Appendix A). Despite the extensive damage observed, there was no indication of lysis in *E. coli* and *L. innocua* after being treated with either LAE or curcumin. This was consistent with previous studies which report that LAE and curcumin alone do not lyse the cells [20,25,26]. The extent of disruption to the cell membrane is also dependent on the fluence of light employed during treatment. An increase in dosage when using methylene blue as a PS has been shown to correspond with a higher number of membrane vesicles or bulges in *E. coli* [27]. Therefore, the treatment using curcumin-LAE solution in this study was sufficient to inactivate bacteria but did not lyse them.

### 3.5. Live/Dead Cell Assay

A live/dead cell assay was conducted to determine the permeability of the bacterial cell membrane before and after treatments. Irradiation of *E. coli* with curcumin at pH 3.5 caused permeation of the membrane (Figure 4). For *L. innocua*, significantly greater portions of cells had a permeable membrane after they were irradiated, which was in line with the cell recovery data (Figure 1, Appendix A). Interestingly, all samples treated with LAE resulted in a large proportion of cells with a permeable membrane regardless of whether the sample was irradiated or not. Therefore, for both *E. coli* and *L. innocua*, it can be concluded that cells with damaged membranes are not necessarily dead based on the cell recovery data and live/dead cell micrographs. Both LAE and curcumin, after irradiation, could damage the membrane, making it more permeable. However, the antimicrobial effectiveness of these treatments depended on the solution pH, as curcumin was more effective at pH 3.5 while LAE was better at pH 7. The reason why no recovery of *E. coli* treated with curcumin-LAE solution at pH 3.5 was observed may have been due to the combined effect of LAE and curcumin, as LAE could damage the cell membrane and allow enhanced interaction between cellular components and ROS produced from irradiated curcumin. This is in line with previous studies, which also reported that disruption in the outer membrane of Gram-negative bacteria using polymyxin nonapeptide or other antimicrobial surfactants increased the antimicrobial efficacy of PSs [15,28]. Proximity between the PS and essential components in the cell membrane is crucial for ROS to exert inhibitory effects. The most stable form of ROS is ^1^O_2_ with a lifetime of 3.5 μs in an aqueous phase, allowing it to travel only tens to hundreds of nanometers [29]. Additionally, ROS have a high reactivity with other non-essential biomolecules, which further emphasizes the importance of PS localization. For instance, a previous study found that the PSs Mg and Zn-tetrabenzoporphyrin have a different photoactivated antimicrobial efficacy despite having the same production yield of ^1^O_2_ [28]. This was attributed to the difference in partitioning properties of these compounds in the cell [28]. Also, the PS hematoporphyrin showed a greater efficacy in binding to the cytoplasmic membrane of *E. coli* following the removal of the outer membrane, leading to increased photoinactivation of the bacteria [30]. This highlights the essential role of the binding of PSs to the cytoplasmic membrane as a requirement for the successful photoinactivation of the bacteria [30].

## 4. Conclusions

The irradiated curcumin-LAE solution had the best antimicrobial activity, as none of the treated bacteria exhibited recovery after the treatment for 24 h at both pH 3.5 and 7 compared to when cells were treated with individual components of the curcumin-LAE solution. The most noticeable difference was observed when *E. coli* was treated with curcumin-LAE at pH 3.5. This was due to the increase in stability of curcumin in water, as the presence of LAE could slow down the crystallization of curcumin even when it was not forming a micelle. Synergistic photoinactivation of LAE and curcumin was observed due to the higher degree of permeation caused by LAE which may have allowed curcumin to localize effectively to the cell membrane and promoted the interaction between cell components and generated ROS as a result of the photosensitization process of irradiated curcumin. This study provides additional insight on how a combination of antimicrobial surfactant and a PS could efficiently inactivate harmful bacteria.

## Figures and Tables

**Figure 1 foods-12-04195-f001:**
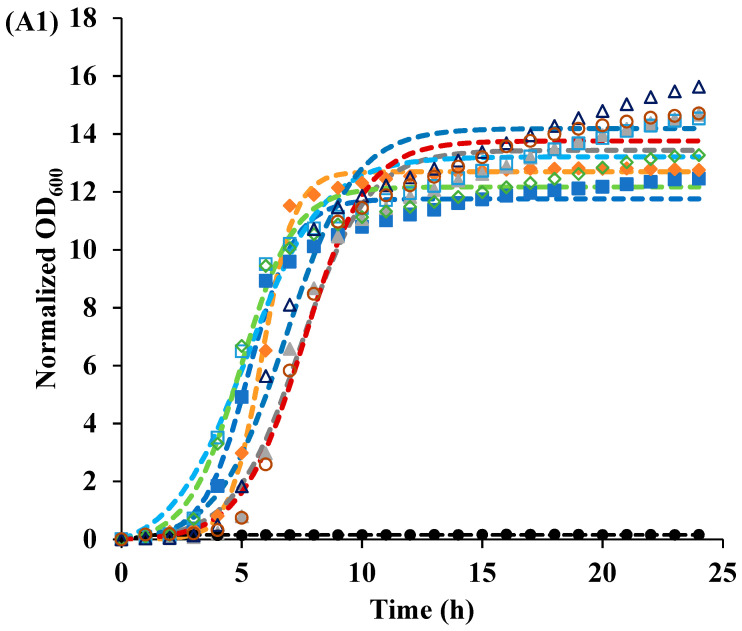
Recovery of the (**A**) *E. coli* and (**B**) *L. innocua* cocktails after irradiation (IR) or non-irradiation (NR) in different solutions at (**1**) pH 3.5 (**A1**,**B1**) and (**2**) pH 7 (**A2**,**B2**). The symbols of samples that were IR (closed) or NR (open) were represented as the control (square), 1 μmol/L curcumin (diamond), 105 μg/mL LAE (triangle), and curcumin-LAE solution (circle).

**Figure 2 foods-12-04195-f002:**
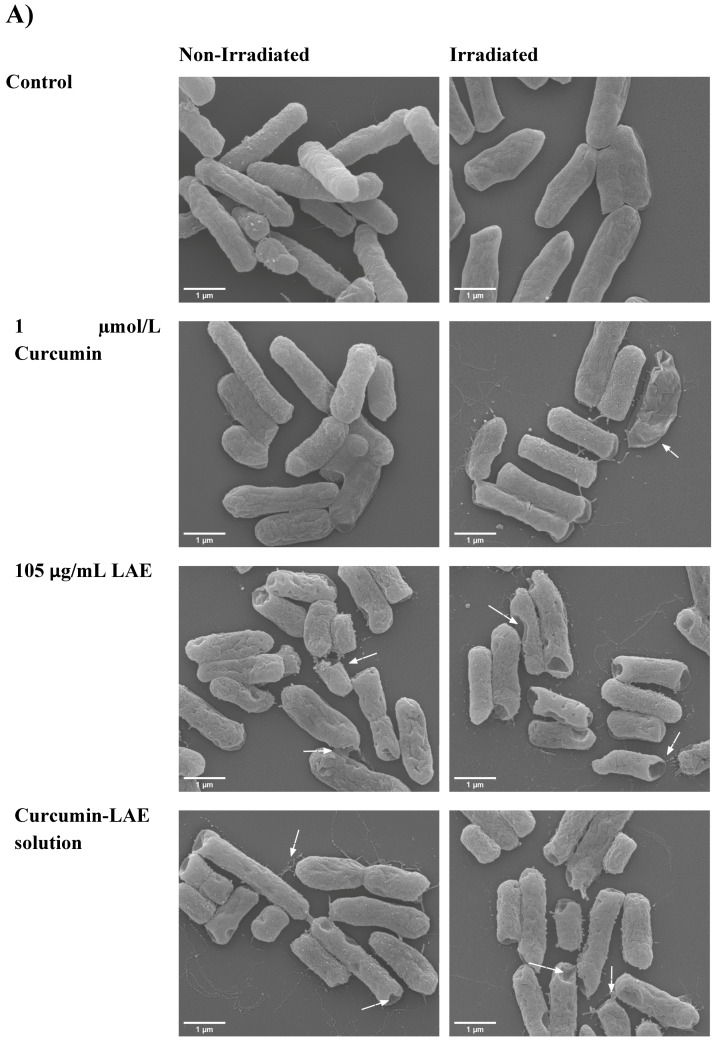
SEM micrograph of a non-irradiated and irradiated cocktail of *E. coli* treated with curcumin, LAE, or curcumin-LAE solutions at (**A**) pH 3.5 and (**B**) pH 7. The white arrows indicate damaged cell surfaces.

**Figure 3 foods-12-04195-f003:**
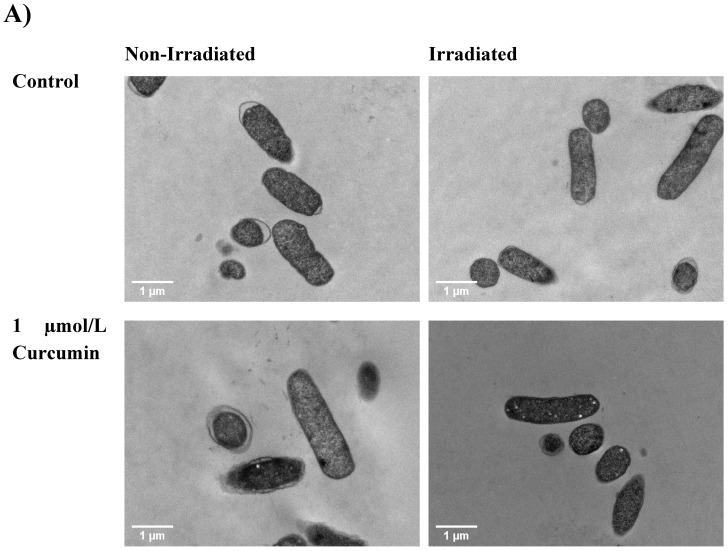
TEM micrograph of a non-irradiated and irradiated cocktail of *E. coli* treated with curcumin, LAE, or curcumin-LAE solutions at (**A**) pH 3.5 and (**B**) pH 7. The white arrows indicate dissolution of the cellular membrane or formation of the membrane channel.

**Figure 4 foods-12-04195-f004:**
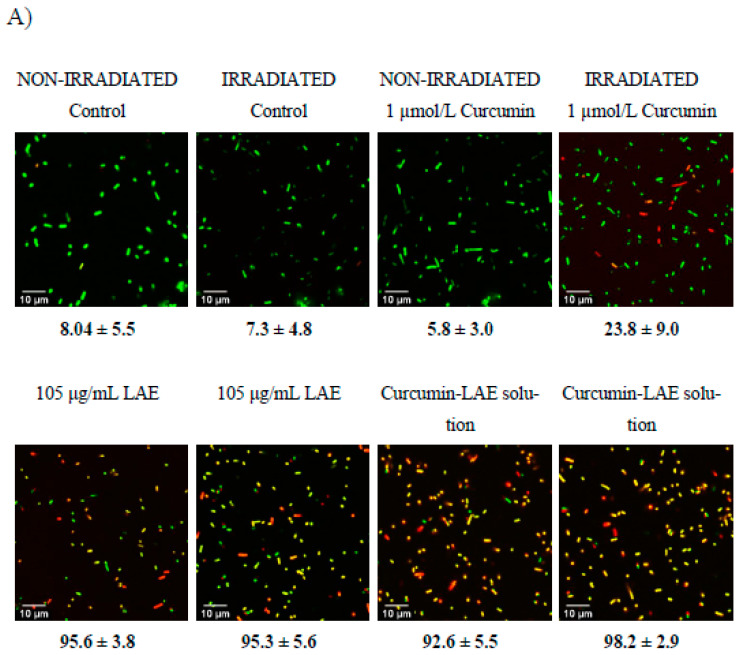
Micrographs of a non-irradiated and irradiated cocktail of *E. coli* treated with curcumin, LAE, or curcumin-LAE solutions at (**A**) pH 3.5 and (**B**) pH 7, stained with Syto9 (green) and propidium iodide (yellow, red). The percentage of the cocktail of *E. coli* at pH 3.5 or pH 7 with permeable membranes (yellow, red) is listed below each micrograph.

**Table 1 foods-12-04195-t001:** Growth parameters of the *E. coli* cocktail treated with curcumin-LAE solutions and their individual components, determined using Equation (1) as a model.

Sample	pH	Irradiated	Parameters		
			*a*	*k*	*t_c_*
			(-)	(h^−1^)	(h)
Control	3.5	NO	13.7	0.63	5.24
		YES	11.8	1.06	5.32
	7	NO	12	0.29	7.08
		YES	12.7	0.28	7.5
1 μmol/L Curcumin	3.5	NO	12.3	0.91	4.99
		YES	12.7	1.51	5.85
	7	NO	8.31	0.48	5.54
		YES	6.06	1.49	5.05
105 μg/mL LAE	3.5	NO	14.3	0.71	6.85
		YES	13.5	0.72	7.46
	7	NO	4.51	1.24	18.2
		YES	0.64	3.07	1.4
Curcumin-LAE solution	3.5	NO	13.8	0.77	7.59
		YES	0.16	5.53	0.63
	7	NO	0.61	1.34	1.54
		YES	0.69	1.17	1.34

**Table 2 foods-12-04195-t002:** Growth parameters of the *L. innocua* cocktail treated with curcumin-LAE solutions and their individual components, determined using Equation (1) as a model.

Sample	pH	Irradiated	Parameters		
			*a*	*k*	*t_c_*
			(-)	(h^−1^)	(h)
Control	3.5	NO	3.41	0.97	8.67
		YES	3.96	0.79	10
	7	NO	4	1.01	8.32
		YES	4.06	0.91	9.51
1 µmol/L Curcumin	3.5	NO	4.24	0.79	9.03
		YES	- ^1^	-	-
	7	NO	4.23	0.74	8.95
		YES	4.34	0.75	18.9
105 μg/mL LAE	3.5	NO	3.98	0.23	17.8
		YES	1.51	0.72	17.8
	7	NO	-	-	-
		YES	-	-	-
Curcumin-LAE solution	3.5	NO	2.66	0.66	17.7
		YES	-	-	-
	7	NO	0.02	1.59	2.47
		YES	0.01	0.25	3.2

^1^ “-“ indicates that the data could not be fitted in the model.

## Data Availability

The data used to support the findings of this study can be made available by the corresponding author upon request.

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
