# Peer review of "Mechanism of Synergistic Photoinactivation Utilizing Curcumin and Lauric Arginate Ethyl Ester against Escherichia coli and Listeria innocua"

_foods, 2023, doi:10.3390/foods12234195_

Round 1
Reviewer 1 Report
Comments and Suggestions for Authors
The authors have studied the mechanism of synergistic photoinactivation utilizing curcumin and LAE against gram-negative E. coli and gram-positive L. innocua. In general, the manuscript has been prepared carefully. Introduction is compact and highlights the important factors, such as photodynamic inactivation, and photosensitizers, as well as the possible health effects of curcumin (in short). The materials and methods are well described and detailed. Results are clearly shown in tables and figures as well as in supplement. The discussion is concise and focuses on the results achieved. Later on it would be interesting to read how the combination of curcumin and LAE work in food, such as a seasoning on top of a chicken fillet, for example, but that is another story.
To consider:
The introduction part starts (l 42-45) with the reference of cancer cells. However, there are more relevant data on food and photodynamic inactivation, so I would recommend staying in this field.
Section 2.4 Bacterial culture conditions: at first, I had the impression that both bacterial species, E. coli and L. innocua were mixed all together as a mixed population. Can you emphasize that they were “separate” stocks by the species?
Minor comments:
Tables 1 and 2: 3.5 should not be in bold (cell control-pH)
line 323: innocua
line 457: not applicable
Author Response
The authors have studied the mechanism of synergistic photoinactivation utilizing curcumin and LAE against gram-negative E. coli and gram-positive L. innocua. In general, the manuscript has been prepared carefully. Introduction is compact and highlights the important factors, such as photodynamic inactivation, and photosensitizers, as well as the possible health effects of curcumin (in short). The materials and methods are well described and detailed. Results are clearly shown in tables and figures as well as in supplement. The discussion is concise and focuses on the results achieved. Later on it would be interesting to read how the combination of curcumin and LAE work in food, such as a seasoning on top of a chicken fillet, for example, but that is another story.
To consider:
The introduction part starts (l 42-45) with the reference of cancer cells. However, there are more relevant data on food and photodynamic inactivation, so I would recommend staying in this field.
Response: The portion of the sentence mentioning about photodynamic therapy to kill cancer cells has been deleted. The following sentence has been deleted:
Line 43: Photodynamic therapy (PDT) uses a combination of light and photosensitizing agent (Rineh et al., 2017)
Section 2.4 Bacterial culture conditions: at first, I had the impression that both bacterial species, E. coli and L. innocua were mixed all together as a mixed population. Can you emphasize that they were “separate” stocks by the species?
Response: Sorry for the confusion, the sentences have been modified to
Line 139: “Two separate cocktails of E. coli or L. innocua were prepared from three strains of E. coli O157:H7 (ATCC-700728), K12 (ATCC-23716), and Seattle 1946 (ATCC-25922), and three strains of L. innocua Seelinger (ATCC-43547, 33090, 51742) purchased from American Type Culture Collections (Manassas, VA, USA).”
Minor comments:
Tables 1 and 2: 3.5 should not be in bold (cell control-pH)
Response: The mentioned 3.5, control, and pH were unbolded (Table 1, 2).
line 323: innocua
Response: typo has been fixed
line 457: not applicable
Response: typo has been fixed
Reviewer 2 Report
Comments and Suggestions for Authors
This is a well done elucidation of the mechanism of action of a new sanitizer the authors are proposing. I have severe limitations as to the applicability of their proposed sanitizer combination any where in the food industry.
I have reservations. I understand that you wanted to determine the mode of action from this proposed sanitizer combination, but I'd like to seen a comparison to the commercially available sanitizers, differences in prices and some conversations with commercial food hygiene personal as to the potential application. I'm hard pressed to see one.
Author Response
This is a well done elucidation of the mechanism of action of a new sanitizer the authors are proposing. I have severe limitations as to the applicability of their proposed sanitizer combination any where in the food industry.
I have reservations. I understand that you wanted to determine the mode of action from this proposed sanitizer combination, but I'd like to seen a comparison to the commercially available sanitizers, differences in prices and some conversations with commercial food hygiene personal as to the potential application. I'm hard pressed to see one.
Response: we understand that this is a study of limited scope. The most probable location that this kind of sanitizer could be applied is food-contact surfaces. However, when comparing with conventional sanitizers, the production cost would be significantly higher for these sanitizers. The authors will be finding other food-grade photosensitizers and compare it to conventional sanitizers in the future study. The current study is to see the mechanism of the formulated sanitizer and forms a base for what we will focus on when formulating photosensitizer-based sanitizer utilizing delivery system.
Reviewer 3 Report
Comments and Suggestions for Authors
REVIEW - foods-2699308
Manuscript titled “Mechanism of synergistic photoinactivation utilizing curcumin and lauric arginate ethyl ester against Escherichia coli and Listeria innocua” raises an interesting topic, i.e. the use of photodynamic therapy (PDT) to combat two widespread bacteria - E. coli and L. innocua (a surrogate of the highly pathogenic L. monocytogenes). An interesting procedure is the use of curcumin in research, a widely researched and popular substance with a wide range of applications for human well-being.
The Introduction briefly and clearly presents the current state of knowledge and logically indicates the purposefulness of the topic discussed. However, it seems to me that the last paragraph, i.e. the goal, was formulated inappropriately by the Authors. The goal should be the goal, not the Authors’ hypotheses or presumptions. There is no need to include any methodology or conclusion in this paragraph.
The Materials and Methods section is unobjectionable.
The Authors obtained interesting results and their graphic presentation is impressive (especially microscopic images). However, it seems to me that their description is very unclear. This part was hard to read. Maybe this can be done in a more accessible way? First with a clear description of the results obtained by the Authors, and after each part of the results and a short discussion? I have the impression that in its current form there is a lot of chaos there. Such results deserve high citations in the future, and for this to happen, the content must be more accessible and inviting to readers.
Author Response
Manuscript titled “Mechanism of synergistic photoinactivation utilizing curcumin and lauric arginate ethyl ester against Escherichia coli and Listeria innocua” raises an interesting topic, i.e. the use of photodynamic therapy (PDT) to combat two widespread bacteria - E. coli and L. innocua (a surrogate of the highly pathogenic L. monocytogenes). An interesting procedure is the use of curcumin in research, a widely researched and popular substance with a wide range of applications for human well-being.
The Introduction briefly and clearly presents the current state of knowledge and logically indicates the purposefulness of the topic discussed. However, it seems to me that the last paragraph, i.e. the goal, was formulated inappropriately by the Authors. The goal should be the goal, not the Authors’ hypotheses or presumptions. There is no need to include any methodology or conclusion in this paragraph.
Response: The authors modified the paragraph by deleting our hypothesis, methodology we employed, and conclusion. “This study investigated the mechanism of synergistic photoinactivation between curcumin and LAE. Our goal was to provide information on potential mechanisms of synergistic antimicrobial activity when both PS and the selected surfactant, i.e., LAE are present.”
The Materials and Methods section is unobjectionable.
The Authors obtained interesting results and their graphic presentation is impressive (especially microscopic images). However, it seems to me that their description is very unclear. This part was hard to read. Maybe this can be done in a more accessible way? First with a clear description of the results obtained by the Authors, and after each part of the results and a short discussion? I have the impression that in its current form there is a lot of chaos there. Such results deserve high citations in the future, and for this to happen, the content must be more accessible and inviting to readers.
Response: The authors agree that the paragraph could be unclear if the results are not stated first. However, we were trying to introduce two perspective results which were from the Growth curve and modeled datasets vs SEM, TEM, and Live/Dead cell assay. For growth curve and modeled datasets, we concluded that the viability of the cell was only affected when both LAE and curcumin are present in the sample at acidic pH for E. coli. However, for SEM, TEM, and Live/Dead cell assay micrograph, we could observe that all cells that were treated with treatment including LAE were damaged. By combining these two results, we concluded that LAE (for both L. innocua and E. coli) or curcumin (for L. innocua) when applied alone causes permeation in the membrane while not affecting the viability of the cell. The viability of the cell was only affected when both LAE and curcumin are present in the sample at acidic pH. Therefore, we tried to put the methodology we’ve employed in front of the paragraph and our findings in the back since it could be confusing as our results are different depending on the methodology used in each section.
The following sentences were revised in order to address the reviewer’s comment of putting the statement about result in front of the sentence for clarification.
Line 248: modified the sentence from
“The dilution of 100 µmol/L stock curcumin-LAE micelle solution to the treatment level of 1 µmol/L curcumin-LAE solution, based on the curcumin concentration, used in this study causes the micelles become destabilized as the critical micelle concentration of LAE is around 5.7 μmol/L (Asker et al., 2009). Even with the destabilization of micelles, the synergistic photoactivated antimicrobial activity was observed between 1 μmol/L curcumin and 105 μg/mL LAE in the previous study (Ryu et al., 2023).”
to
“In the previous study, the synergistic photoactivated antimicrobial activity was observed between 1 μmol/L curcumin and 0.248 µmol/L LAE (Ryu et al., 2023). The dilution of 100 µmol/L stock curcumin-LAE micelle solution to the treatment level of 1 µmol/L curcumin-LAE solution, based on the curcumin concentration, destabilized micelles as the concentration of LAE present in the diluted solution is 0.248 µmol/L while LAE’s critical micelle concentration is around 5.7 μmol/L (Asker et al., 2009). Therefore, even with the destabilization of the LAE micelle, synergistic antimicrobial activity between curcumin and LAE was observed.”
Line 322: Cocktail of E. coli was more resistant toward treatment compared to the L. innocua cocktail.
Reviewer 4 Report
Comments and Suggestions for Authors
The manuscript describes an interesting study aimed at investigating the bacterial inhibition mechanism through photoinactivity. English is good, the methodology is adequate and the results are properly shown. Only a few formal observations can be made, which are detailed:
-The genus and species of microorganisms must always be written in Italics. Check References
- Title: If only the genus and species of the used miroorganisms are written, it can be understood that all the strains that belong to them have the same behavior. As is known, there is dependence strain so that in the title "strains" after "Listeria innocua" must be added. Idem in line 30
- When several references are cited together, it is convenient to write them in chronological order. Ex: see Line 67
Author Response
The manuscript describes an interesting study aimed at investigating the bacterial inhibition mechanism through photoinactivity. English is good, the methodology is adequate and the results are properly shown. Only a few formal observations can be made, which are detailed:
-The genus and species of microorganisms must always be written in Italics. Check References
Response: The authors used endnote for references. However, there seems to be a conversion error. The genus and species in the reference have been italicized.
- Title: If only the genus and species of the used miroorganisms are written, it can be understood that all the strains that belong to them have the same behavior. As is known, there is dependence strain so that in the title "strains" after "Listeria innocua" must be added. Idem in line 30
Response: We agree with the reviewer that the response of a specific microorganism may depend on strains. Therefore, we used a cocktail of three strains of Listeria innocua. The strains have been specified in the Materials and Methods section.
Line 31: The word “strains” was added.
- When several references are cited together, it is convenient to write them in chronological order. Ex: see Line 67
Response: The authors used APA 6th edition format on endnote, and according to them It says that we should cite in alphabetical order and separate them with semicolons. If two or more works have the same author, it should be arranged by year of publication.
Round 2
Reviewer 2 Report
Comments and Suggestions for Authors
Thank you for understanding and agreeing that spraying curcumin all over food contact surfaces and then using UV-A light as a combined sanitizer has significant limitations! Was there commercial interest in your Ryu, 2023 published data? Add this to L 250.
You have made the changes suggested by the three reviewers.
I’d like to see a fuller explanation of the limitations we’ve agreed to in the conclusions, but I will not insist on this.
Future consideration use UV C rather than UV A—it’s much more lethal to microorganisms!